# Effect of alteplase, benzodiazepines and beta-blocker on post-stroke pneumonia: Exploration of VISTA-Acute

Thanh G. Phan[1,2]*, Richard Beare[3,4], Philip M. Bath[5], Svitlana Ievlieva[1,2], Stella Ho[6], John Ly[1,2], Amanda G. Thrift[2], Velandai K. Srikanth[3], Henry Ma[1,2], on behalf of the VISTA-Acute Collaborators[¶]

1 Department of Neurology, Monash Medical Centre, Clayton, Australia, 2 Stroke and Aging Research Group, School of Clinical Sciences at Monash Health, Monash University, Melbourne, Australia, 3 Department of Medicine, Peninsula Health and Central Clinical School, Monash University and National Centre for Healthy Ageing, Melbourne, Australia, 4 Murdoch Children Institute of Research, Melbourne, Australia, 5 Division of Clinical Neuroscience, Stroke Trials Unit, University of Nottingham, Nottingham, United Kingdom, 6 Department of Pharmacy, Monash Health, Clayton, Australia

¶ Complete membership of the author group can be found in the Acknowledgments
* Thanh.Phan@monash.edu

**Data Availability Statement:** The VISTA-Acute data were pooled from de-identified randomized control trials. The governance of the registry can be found at https://www.virtualtrialsarchives.org/

## Abstract

### Background

Post-stroke pneumonia is a frequent complication of stroke and is associated with high mortality. Investigators have described its associations with beta-blocker. However, there has been no evaluation of the role of recombinant tissue plasminogen activator (RTPA). We postulate that RTPA may modify the effect of stroke on pneumonia by reducing stroke disability. We explore this using data from neuroprotection trials in Virtual International Stroke Trials Archive (VISTA)-Acute.

### Method

We evaluated the impact of RTPA and other medications in random forest model. Random forest is a type of supervised ensemble tree-based machine learning method. We used the standard approach for performing random forest and partitioned the data into training (70%) and validation (30%) sets. This action enabled to the model developed on training data to be evaluated in the validation data. We borrowed idea from Coalition Game Theory on fair distribution of marginal profit (Shapley value) to determine proportional contribution of a covariate to the model. Consistent with other analysis using the VISTA-Acute data, the diagnosis of post-stroke pneumonia was based on reports of serious adverse events.

### Results

The overall frequency of pneumonia was 10.9% (614/5652). It was present in 11.5% of the RTPA (270/2358) and 10.4% of the no RTPA groups (344/3295). There was significant (p<0.05) imbalance in covariates (age, baseline National Institutes of Health Stroke Scale (NIHSS), diabetes, and sex). The AUC for training data was 0.70 (95% CI 0.65–0.76),

vista/. The analysis of the results of this study was performed on virtual network computer provided by VISTA-Acute. As such the authors cannot provide the data. However, the data used in this study is available on request from VISTA-Acute, who owns the data. The authors confirm that others would be able to access these data in the same manner. The authors did not have any special access privileges that others would not have.

**Funding:** This study is not funded. The funders had no role in study design, data collection and analysis, decision to publish, or preparation of the manuscript.

validation data was 0.67 (95% CI 0.62–0.73). The Shapley value shows that baseline NIHSS ($\geq$10) and age ($\geq$80) made the largest contribution to the model of pneumonia while absence of benzodiazepine may protect against pneumonia. RTPA and beta-blocker had very low effect on frequency of pneumonia.

## Conclusion

In this cohort pneumonia was strongly associated with stroke severity and age whereas RTPA had a much lower effect. An intriguing finding is a possible association between benzodiazepine and pneumonia but this requires further evaluation.

## Introduction

Stroke is a leading cause of disability worldwide and results in significant economic and societal cost [1]. Patients who have post-stroke pneumonia have higher mortality and disability than those patients without this complication [2]. Investigators have reported a 10% prevalence of post-stroke pneumonia [2]. The frequency of pneumonia is higher among patients with low level consciousness or among those admitted to intensive care [3]. Several different approaches have been employed in trial settings to prevent post-stroke pneumonia. These approaches, such as prophylactic administration of antibiotics and use of nasogastric tube feeding, have not been successful [2,4–6]. There is a suggestion that metoclopramide may reduce the risk of post-stroke pneumonia among patients who are fed using a nasogastric tube but such finding was confined to one report [7].

Using data from from Virtual International Stroke Trials Archive (VISTA)-Acute, investigators have reported associations between beta-blockers, but not benzodiazepines, and post-stroke pneumonia [8,9]. Beta-blockers had been postulated to have an effect by dampening activation of the sympathetic pathway. The sympathetic pathway has been postulated to be activated as part of immune dysfunction stroke [10]. Recombinant tissue plasminogen activator (RTPA) on pneumonia has been shown to reduce stroke severity. The strong relationship between stroke severity and pneumonia prompted us to consider the possibility that reversal of stroke severity such as by RTPA can decrease the risk of pneumonia. In our previous analysis of data from a single centre, we had observed only 4/85 (4.7%) cases of pneumonia among patients who received RTPA compared with higher frequency of pneumonia (6.6%) in the entire cohort [11]. In the larger VISTA-Acute cohort, we aimed to evaluate the effect of RTPA on development of post-stroke pneumonia taking advantage of information available in this archive on different medications such as sleeping medications, antipsychotic drugs [8,9]. These medications were included as they are centrally acting and can impact on level of consciousness. We explore a model of post-stroke pneumonia using the new development of interpretable machine learning to make the findings from random forest accessible [12–14].

## Method

We used data from the Virtual International Stroke Trials Archive (VISTA)-Acute archives of stroke clinical trials [15]. This data pertains to neuroprotection trials rather than recent RTPA or thrombectomy trials. The VISTA-Acute data were pooled from de-identified randomized control trials. The governance of the registry can be found at https://www.virtualtrialsarchives.org/vista/. The analysis of the results of this study was performed on virtual network computer

provided by VISTA-Acute. As such the authors cannot provide the data. However, the data used in this study is available on request from VISTA-Acute, who owns the data. The authors confirm that others would be able to access these data in the same manner. The authors did not have any special access privileges that others would not have.

The following terms were used in the search strategy of this dataset: imaging data; National Institutes of Health Stroke Scale (NIHSS) on admission and at 24 hours; physiological variables (systolic blood pressure, blood glucose level); demographic data (age, sex); stroke risk factors; thrombolysis treatment with recombinant tissue plasminogen activator (RTPA); -Alberta Stroke Program Early CT Score (ASPECTS) score on CT scan [the ASPECTS assesses the extent of ischemia over 10 regions of the middle cerebral artery territory] [16]; medication data (drug class such as beta-blockers (BB2), antipsychotics (PSYCH2), benzodiazepines (BENZ2), sleeping aid medications (SLEE2)); clinical data (atrial fibrillation (AFIB), heart failure (CHF), ischemic heart disease (IHD), diabetes, hypertension); and modified Rankin outcome within 90 days of stroke. A Rankin score of 0 signifies no symptom while 6 signifies death. In this study disability is defined by a Rankin score >2.

Consistent with previous works from VISTA-Acute on this subject, diagnosis of post-stroke pneumonia was based on reports of serious adverse event within 10 days of onset [8]. We had used a broad collection of terms to assign the diagnosis of pneumonia. These terms include bronchopneumonia but not upper tract infection. We evaluate medications which the patients are taking. We extract the drug class from the list of medications that the patient was on. This data is part of the metadata acquired during drug trial. These drugs are stored with their Anatomical Therapeutic Chemical Classification System (ATC code) within VISTA. These drugs such as beta-blockers were not the the subject of the trial but were drugs prescribed by clinicians in daily management. VISTA-Acute does not include data on the type of dysphagia screen used, details about mechanical ventilation nor evaluation of immune dysfunction.

Steps in machine learning to identifed features important to post-stroke pneumonia is listed below.

1-Search text for terms to diagnose pneumonia

2-Partition data to training and validation

3-Perform Random Forest on training dataset

4-Evaluate the model again on validation dataset

5-Asssess importance of identified features using Shapley value

## Statistical analysis- machine learning

We used random forest, a supervised machine learning method related to decision tree analysis, which employed random selection of covariates and patients from the dataset to create multiple trees (S1 Fig). This form of ensemble learning utilises the average outcome of multiple trees (n = 200) or 'wisdom of the crowd' to create the final model. The analysis was performed using *randomForest* package in R statistical programming environment. Given the interest in finding out the covariates making largest contribution to pneumonia, we calculated the marginal contribution of each covariate to the model or Shapley value (phi) [13,17]. The Shapley value analysis is based on the Nobel prize winning works (Economics) by Lloyd Shapley on Coalition (Cooperative) Game Theory. The marginal contribution is determined as the average of all permutations of the coalition of the covariates containing the covariate of interest minus the coalition without the covariate of interest. We used the feature importance analyses

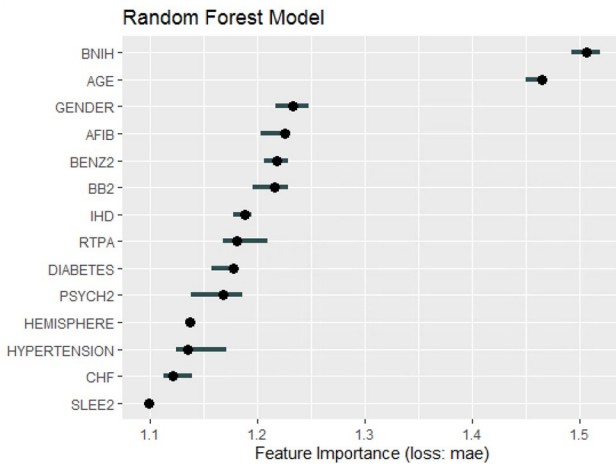
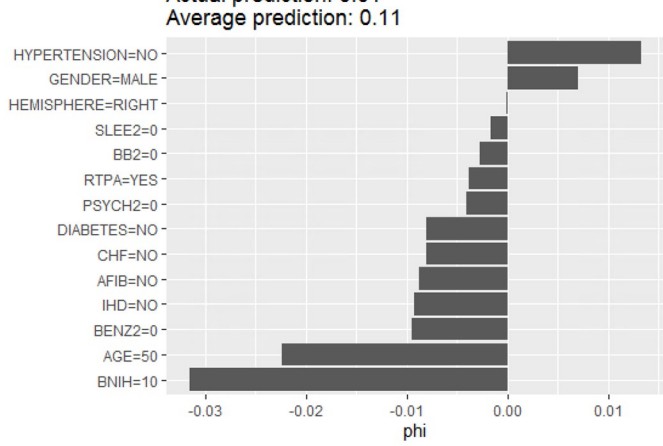

**Fig 1. Shapley value (phi) and feature importance provide different views on the model.** Shapley value illustrates the 'payout' of the feature within the model and the direction of its impact on the classification of pneumonia. The difference between the actual and average prediction implies the sum of the payout for prediction is 0.10. Benzodiazephine has a larger payout that alteplase (RTPA) or beta-blockers on pneumonia. Feature importance illustrates the impact on the model's error by permuting the features. BNIH = baseline NIHSS, BENZ2 = benzodiazepine, BB2 = beta blockers, SLEE2 = non-benzodiazepine sleeping medications, IHD = ischemic heart disease, CHF, congestive heart failure, AFIB = atrial fibrillation, RTPA = recombinant tissue plasminogen activator.

to assess the impact of permuting (re-ordering) the feature (covariate) on the model's prediction error (Fig 1). The individual conditional expectation (ICE) curve describes the effect of the individual observation (e.g. baseline NIHSS) on the outcome (probability of pneumonia, Fig 2) [18]. Next the interaction strength of the feature is measured using H-statistic (see S2 Fig for example of interaction strength with covariate) [12]. The measure how much of the variation of the prediction of the feature depends on the interaction with other features. The H-statistic is 0 if there is no interaction and 1 if the effect of the prediction comes from the interaction. This analysis was performed using *iml* package in R [14].

We used the standard approach in machine learning, partitioning the data into training (70%) and validation (30%) sets. The model was developed using the training sample, and then the very same model is applied to the validation sample to evaluate if the model achieves the same discrimination result. We test the model in several ways including measuring the AUC and the out of bag error. The discrimination of the model is measured by area under Receiver Operating Characteristics (ROC) curve (AUC). An AUC of 0.50–0.59 indicates that it is no better than chance, 0.60–0.69 indicates poor discrimination, 0.70–0.79 indicates fair discrimination; 0.80–0.89 indicates good discrimination and >0.90 indicates excellent discrimination.

## Results

In total, 5653 eligible patients were identified, with 3087 (54.6%) male, median age 71 years (IQR 54–88), NIHSS 12 (IQR 3–21), RTPA 41.7% (see Table 1). The overall frequency of pneumonia was 10.9% (614/5652). It was present in 11.5% of the RTPA (270/2358) and 10.4% (344/3295) of the no RTPA groups. 614 (10.7%) cases. The two groups differed significantly with regards to age (p<0.001), baseline NIHSS (p<0.001), beta blockers (p<0.001), benzodiazepines (p<0.001), diabetes (p<0.001), hypertension (p<0.001) and sex (p = 0.0049).

### Random forest

The AUC in the training model was 0.70 (95% CI 0.65–0.76), and in the test model was 0.67 (95% CI 0.62–0.73) and out of bag (left over data from training) error for the training data was

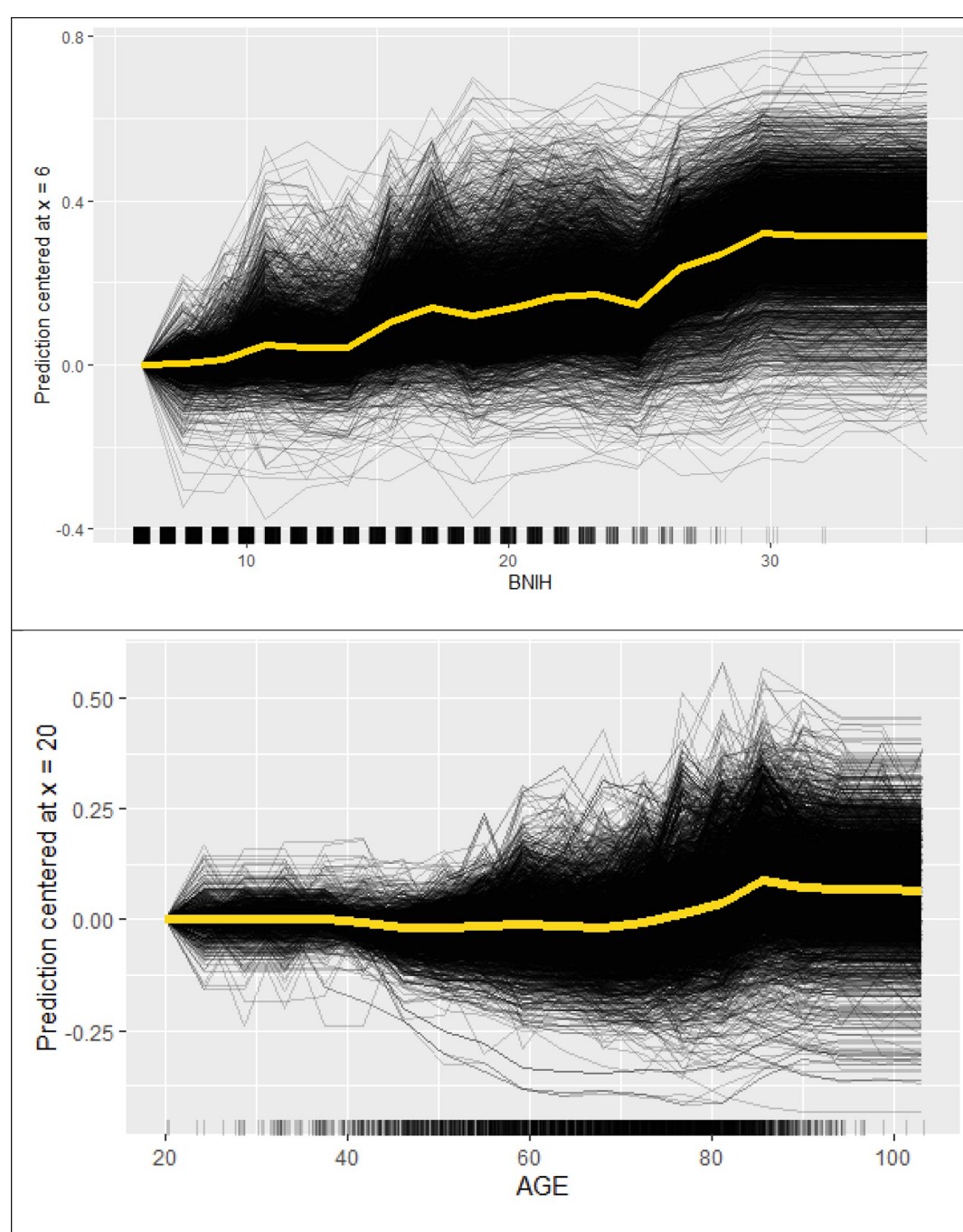

**Fig 2. The lines represents the prediction of pneumonia each patient's baseline NIHSS.** The yellow line is the average of the lines for all patients. The dense black squares on the x-axis represent regions with data and regions from high 20s onward indicate that the data is less strong. The probability of pneumonia increased from tne age of 80 on wards.

11.04% (calculated from left over data to estimate error in model). The Shapley value shows that low baseline stroke severity (NIHSS ≤10), younger age (≤50 years) made the largest contribution to lowering the frequency of pneumonia while absence of history of hypertension and male sex have the largest effect on pneumonia (Fig 1). In this figure, the Shapley value illustrates the 'payout' of the feature within the model and the direction of its impact on the

**Table 1. Comparisons of people receiving RTPA and not receiving RTPA in the VISTA-Acute data set.**

| Features | RTPA (n = 2358) | No RTPA (n = 3295) | p-value |
|---|---|---|---|
| BNIH | 14 (IQR 6–22) | 12 (IQR 4–20) | <0.001 |
| Age | 71 (IQR 53–89) | 72 (IQR 55–89) | <0.001 |
| Sex (%, Male) | 56.8 | 53.2 | 0.0049 |
| BB2 | 0.53 | 0.45 | <0.001 |
| BENZ2 | 0.35 | 0.29 | <0.001 |
| Diabetes | 0.2 | 0.25 | <0.001 |
| Hypertension | 0.7 | 0.76 | <0.001 |
| Atrial Fibrillation | 0.24 | 0.27 | 0.0089 |
| CHF | 0.09 | 0.09 | 0.98 |
| Pneumonia | 0.115 | 0.104 | 0.23 |
| Rankin ≤2 | 0.422 | 0.402 | 0.141 |
| mortality | 0.153 | 0.173 | 0.038 |

BNIH = baseline NIHSS, BENZ2 = benzodiazepine, BB2 = beta blockers, IHD = ischemic heart disease, CHF, congestive heart failure, AFIB = atrial fibrillation, RTPA = recombinant tissue plasminogen activator.

classification of pneumonia. The difference between the actual and average prediction implies that the sum of the payout for prediction is 0.10. Among the medications, alteplase and beta-blockers has minimal effect on pneumonia. By contrast, absence of benzodiazepine has a greater payout on pneumonia. Feature importance illustrates the impact on the model's error by permuting the features. Permutation of age and baseline NIHSS had the greatest effect on the model's error, follow by gender, atrial fibrillation and benzodiazepines and beta blockers. The probability of pneumonia increased in step-wise manner above an NIHSS score of 10 (Fig 2). For age, the probability of pneumonia increased from 80 years of age onwards. We used the interaction strength of the features to illustrate the complex relationship among the features. The interaction strength (H-statistic) is strong for age and baseline stroke severity and lower for RTPA.

## Discussion

The main findings in our study are that RTPA and beta-blockers had a minimal effect on the proportion of people having pneumonia following stroke. Using Shapley value, the baseline stroke severity and age dominate the 'payout' for pneumonia with the probability of pneumonia lower at stroke severity less than 10 and age less than 50. This finding on stroke severity and age is consistent with other studies on stroke-associated pneumonia. Our finding on the possible role of benzodiazepines requires further evaluation in the future.

### VISTA-Acute

The motivation for this study were the potential effect of alteplase on lowering stroke severity [19]. This finding generates the hypothesis that lower NIHSS from RTPA would be associated with lower frequency of pneumonia. However, if the effect of RTPA on pneumonia exists, it's relatively modest as can be visualised from the model. Random forest is based on decision trees but it's not simple to print 200 trees for the readers to understand. This is in part due to the ensemble machine learning process hinted in the introduction whereby the algorithm takes the 'wisdom of the crowd' and average the contribution of the covariate to the model to generate the final model. Machine learning has a reputation as a 'black box' and so has previously been avoided in medicine. Recently, there has been renewed interest in machine learning

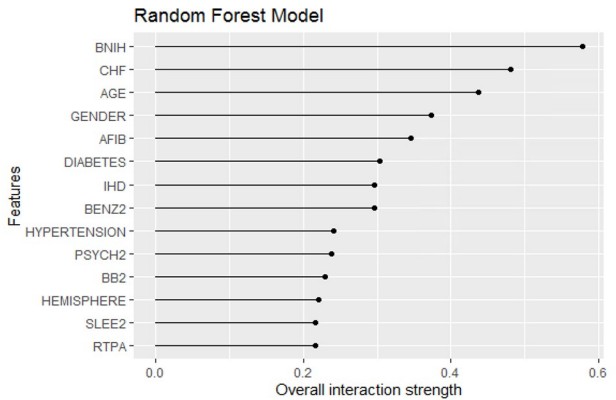
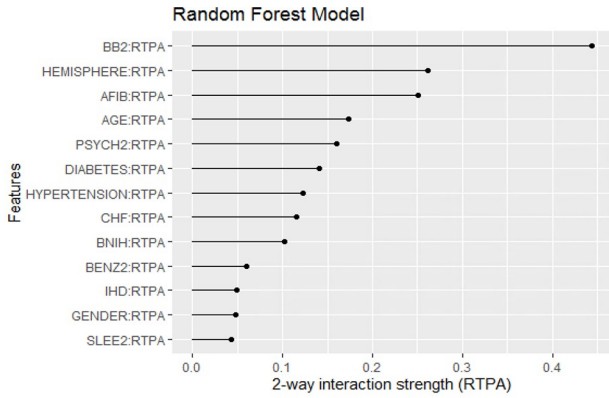
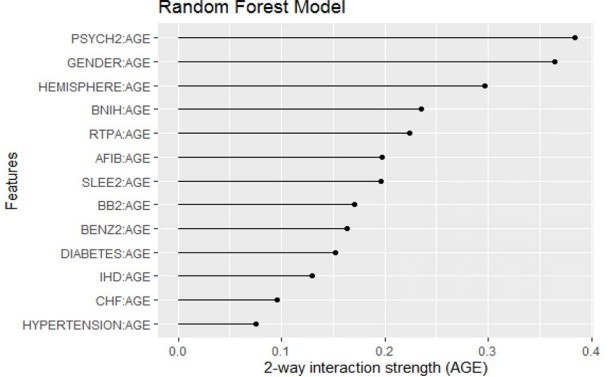
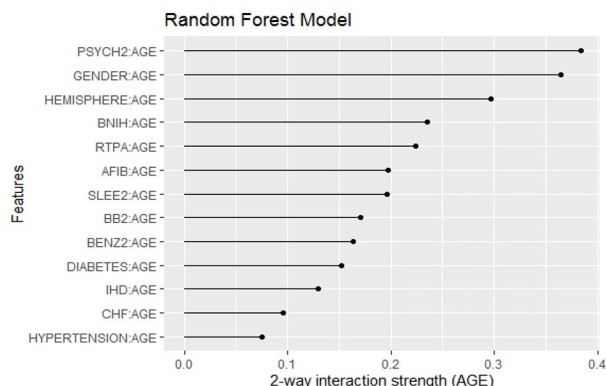

**Fig 3. The interaction strength (H-statistic) is strong for age and baseline stroke severity and almost reach 70% of variance explained per feature.**
BNIH = baseline NIHSS, BENZ2 = benzodiazepine, BB2 = beta blockers, SLEE2 = benzodiazepine and non-benzodiazepine sleeping medications, IHD = ischemic heart disease, CHF, congestive heart failure, AFIB = atrial fibrillation, RTPA = recombinant tissue plasminogen activator.

but publications on the topic has focussed on the outcome model rather than explanation of the model [20]. We took advantage of changes in interpretable machine learning to look inside the 'black box' [14]. This approach permits a range of tools such as Game Theory [13] to determine which covariates make the largest contribution to the model (see Shapley value plot in Fig 1), visualization the relationship between the covariate of interest and the predicted outcome (see ICE plot in Fig 2) [18] and exploring the plot of interaction (Fig 3). These approachs permit examination of multiple covariates whereas more traditional regression approaches have explored the effect of medications on stroke outcome evaluate each medication on its own [8,9]. This approach is different from that obtained by using logistic regression which reduce the covariates down to those with statistical significant association with the variables (age, sex, diabetes and baseline NIHSS). However, this approach does not offer the flexibility of our approach. Within the random forest model it is possible to analyse the interactions among the covariate or the effect of covariate threshold at which the risk of pneumonia increases (see Fig 2). The feature importance plot from random forest ranks diabetes as the ninth in terms of importance whereas the the logistic regression model might draw attention to the p-value and odds ratio (see Table 1).

The key findings from random forest is the importance of baseline NIHSS followed by age and sex (feature importance plot). In this plot, hypertension, heart failure and sleeping medication ranked lowest. The Shapley value plot illustrates the 'payout' of the feature and show the

impact of the feature on the classification of pneumonia in Fig 1. The baseline NIHSS has the largest payout with the threshold being at 10. As the NIHSS increases above 10 the effect on the payout favour pneumonia. This was also the age with threshold at 50-year-old.

These threshold are shown again in the Individual Condition Expectation (ICE) curve in Fig 2. The direction of the payout for hypertension is that the absence of history of hypertension favours classification of penumonia. As the reviewer points out, this finding seems counterintuitative. It's not clear from the data the reason for this. The interaction plot suggests strong interaction with beta-blockers (S2 Fig) but weak interaction between hypertension with age and baseline NIHSS. For example, studies have explored the use of angiotensin converting ezyme inhibitor in reduction of post-stroke pneumonia [21]. However, the data on this is still early and we would caution the interpretation until further data become available.

Our finding on the importance of NIHSS in stroke-associated pneumonia has been demonstrated by other investigators as well as data from our centre [11,22,23]. In our data stroke severity made the largest contributor to pneumonia at 72.8% whereas dysphagia only contributed 3.8% [11]. A limitation is that we did not have data on dysphagia in this clinical trial registry cohort to evaluate its importance. Investigators in the past and recently have alluded to the importance of dysphagia [23]. In analysis of data from our centre and using Shapley value regression, we showed that the proportional contribution of dysphagia to stroke associated pneumonia was small relative to NIHSS [11]. Similar to dysphagia, other important variables such as nasogastric tubes were not available [6]. One possibility is the variability in the management.

## Medications

The prior papers describing a potential effect of beta-blocker [8] on pneumonia but have not explored the effect of other medications such as benzodiazepine in the same model [9]. Our finding suggests a small effect of benzodiazepine on pneumonia. The plot shows that benzodiazepine has a larger payout that alteplase (RTPA) or beta-blockers on pneumonia. The model framed it in the negative as pneumonia was less likely to occur among patients not taking benzodiazepine. A previous analysis of the VISTA-Acute data suggested that benzodiazepine is not associated with post-stroke pneumonia [9]. However, these papers have not analyzed these medications together such as we have done here with random forest [8,9]. Our finding has support in that benzodiazepine has been associated with pneumonia in chronic phase of stroke [24]. Meta-analysis of observation studies on the use of benzodiazepine, outside the setting of stroke, suggested that its use increased the risk of pneumonia [25]. The reason for this is observation not clear. In the context of stroke, benzodiazepine may lower the level of consciousness, but the interaction plot (Fig 3) does not support this postulate. Another possibility is that benzodiazepine may be used as a sedating drug in elderly patients with delirium. Against this postulate, other sleeping aid medications and psychotropic drugs were not associated with increased risk of pneumonia in our cohort.

We do not have data on why these patients were given benzodiazepine and at which stage of the disease that the patients were given benzodiazepine. Because we had examined this drug as a class effect, we did not take dosage into consideration. Furthermore, we cannot be certain if benzodiazepine could have modified the risk of pneumonia in patients who were drowsy, had severe stroke deficit and dysphagia. The role of benzodiazepine in stroke is interesting. Previously, benzodiazepine had been explored as a drug to reduce stroke disability via its action on gamma-amino-butyric acid (GABA) [26]. Others have also proposed that benzodiazepine be used to diagnosed transient ischemic attack via its ability to reintroduce neurological deficit [27]. It is likely that further works need to be done in this area.

## Limitations

A potential issue with using VISTA-Acute data is that the data represents patients randomized to clinical trials and thus the cohort may not reflect the breadth of patients usually seen in hospitals. Those excluded from clinical trials can include individuals with the most severe strokes or have multiple comorbidities. Furthermore, we have data on onset to treatment with neuroprotection drug but do not have raw data on time of onset or time of hospital arrival. Such data can help to determine if pneumonia was related to stroke occurring during sleep. The diagnosis of post-stroke pneumonia was based on reports of serious adverse events [8]. This approach has been used by several different groups evaluating diffferent aspects of pneumonia from VISTA-Acute [8,9,28]. This approach is different from the clinical approach and thus has the potential to under report the number of cases. We were careful in this analysis by having a set dictionary of terms to search for the diagnosis of pneumonia from SAE log. A potential approach is to use ICD-10 codes for the diagnosis of pneumonia but this route was not available given that the data came from clinical trial registry and which contains de-identified data from multiple trials. In this analysis, data on other parameters such as blood test results could not be used due to large proportion of missing values. On the question of thrombectomy, we have not measured this as the trials gathered in this dataset were neuroprotection trials and not RTPA or thrombectomy trials. Since the data were from the pre-thrombectomy era, it is likely that the number of such cases were very small. On the issue of beta-blocker and pneumonia, we found that the proportional contribution of it to the random forest model was low (Fig 1). Our finding appears on the surface different to previous analyis which had used Poisson regression but this might relate to methodological approach [8]. Those investigators had found risk reduction with beta-blockers but with wide confidence interval [8].

## Conclusion

Ensemble machine learning method showed the importance of baseline stroke severity and age in model of post stroke pneumonia. There is a possibility that benzodiazepine may contribute but this needs to be evaluated outside of VISTA type of dataset. More research should be directed at the effect of other medications on stroke outcome. We were not able to confirm the importance of RTPA or beta blocker in development of pneumonia in VISTA-Acute. We will approach other investigators regarding availability of data from thrombolysis trials to perform this analysis prior to concluding the role of RTPA on pneumonia.

## Supporting information

**S1 Fig. The error rate decreases with more trees added.**
(DOCX)

**S2 Fig. The interaction strength (H-statistic) is strong for age and baseline stroke severity and almost reach 70% of variance explained per feature.** BNIH = baseline NIHSS, BENZ2 = benzodiazepine, BB2 = beta blockers, SLEE2 = benzodiazepine and non-benzodiazepine sleeping medications, IHD = ischemic heart disease, CHF, congestive heart failure, AFIB = atrial fibrillation, RTPA = recombinant tissue plasminogen activator.
(DOCX)

**S1 Appendix.**
(DOCX)

## Acknowledgments

Collaborators GBDS. Global, regional, and national burden of stroke and its risk factors, 1990–2019: a systematic analysis for the Global Burden of Disease Study 2019. Lancet Neurol. 2021;20(10):795–820. Epub 2021/09/07. doi: 10.1016/S1474-4422(21)00252-0. PubMed PMID: 34487721; PubMed Central PMCID: PMCPMC8443449 PreventS web app and free Stroke Riskometer app, which are owned and copyrighted by Auckland University of Technology, New Zealand. V Feigin reports grants received from the Brain Research New Zealand Centre of Research Excellence (16/STH/36), National Health & Medical Research Council (NHMRC, Australia APP1182071) and World Stroke Organization to their institution; leadership or fiduciary role in board, society, committee or advocacy group, paid or unpaid with World Stroke Organization as Executive Committee member, New Zealand Stroke Education (charitable) Trust as CEO, Stroke Central New Zealand as Honorary Medical Director, all of which are honorary unpaid roles; all outside the submitted work. O Adebayo reports grants or contracts from Merck Foundation; support for attending meetings and/or travel from Novartis; all outside the submitted work. R Akinyemi reports grants or contracts from NIH (U01HG010273), and GCRF (GCRFNGR6\1498), all outside the submitted work. R Ancuceanu consulting fees from AbbVie and AstraZeneca; payment or honoraria for lectures, presentations, speakers bureaus, manuscript writing or educational events from AbbVie, Sandoz, and B. Braun; support for attending meetings and/or travel from AbbVie and AstraZeneca; all outside the submitted work. J Arnlov reports payment or honoraria for lectures, presentations, speakers bureaus, manuscript writing or educational events from AstraZeneca and Novartis; participation on a Data Safety Monitoring Board or Advisory Board with AstraZeneca and Boehringer Ingelheim; all outside the submitted work. Z Aryan reports support for the present manuscript from American Heart Association as funding to their institution, and from Brigham and Women's Hospital as an employee. M Ausloos reports grants or contracts from [Romanian National Authority for Scientific Research and Innovation, CNDS-UEFISCDI project number PN-III-P4-ID-PCCF-2016-0084, research grant (Oct 2018-Sept 2022), grant title "Understanding and modelling time-space patterns of psychology-related inequalities and polarization" outside the submitted work. T Barnighausen reports grants or contracts from Research grants from the European Union (Horizon 2020 and EIT Health), German Research Foundation (DFG), US National Institutes of Health, German Ministry of Education Research, Alexander von Humboldt Foundation, Else-Kroner-Fresenius-Foundation, Wellcome Trust, Bill & Melinda Gates Foundation, KfW, UNAIDS, and WHO; consulting fees from KfW on the OSCAR initiative in Vietnam; participation on a Data Safety Monitoring Board or Advisory Board with the NIH-funded study "Healthy Options" (PIs: Smith Fawzi, Kaaya), Chair of the Data Safety and Monitoring Board (DSMB), German National Committee on the "Future of Public Health Research and Education", Chair of the scientific advisory board to the EDCTP Evaluation, Member of the UNAIDS Evaluation Expert Advisory Committee, National Institutes of Health Study Section Member on Population and Public Health Approaches to HIV/AIDS (PPAH), US National Academies of Sciences, Engineering, and Medicine's Committee for the "Evaluation of Human Resources for Health in the Republic of Rwanda under the President's Emergency Plan for AIDS Relief (PEPFAR)", and University of Pennsylvania (UPenn) Population Aging Research Center (PARC) External Advisory Board Member; leadership or fiduciary role in board, society, committee or advocacy group, paid or unpaid as the co-chair of the Global Health Hub Germany (which was initiated by the German Ministry of Health); all outside the submitted work. E Beghi reports grants or contracts from Italian Health Ministry, American ALS Association, and SOBI Pharmaceutical Company made to their institution; payment or honoraria for lectures, presentations, speakers bureaus, manuscript writing or

educational events from University of Rochester; support for attending meetings and/or travel from ILAE; participation on a Data Safety Monitoring Board or Advisory Board with Arvelle Therapeutics; all outside the submitted work. Y Bejot reports payment or honoraria for lectures, presentations, speakers bureaus, manuscript writing or educational events from Medtronic, Boehringer-Ingelheim, Pfizer, BMS, Servier, and Amgen; support for attending meetings and/or travel from Servier; all outside the submitted work. A Catapano reports grants or contracts from Sanofi, Eli Lilly, Mylan, Sanofi Regeneron, Menarini, and Amgen; payment or honoraria for lectures, presentations, speakers bureaus, manuscript writing or educational events from Akcea, Amgen, AstraZeneca, Aegerion, Amryt, Daiichi, Sankyo, Esperion, Kowa, Ionis Pharmaceuticals, Mylan, Merck, Menarini, Novartis, Recordati, Regeneron, Sandoz, and Sanofi; all outside the submitted work. S Costanzo reports grants or contracts from ERAB (the European Foundation for Alcohol Research; id. EA1767; 2018–2020 and Italian Ministry of Health (grant RF-2018-12367074, CoPI), both paid to their institution; payment or honoraria for lectures, presentations, speakers bureaus, manuscript writing or educational events from as a member of the Organizing Committee and speaker for the 9th European Beer and Health Symposium (Bruxelles 2019) and for given lecture at the 13th European Nutrition Conference FENS 2019 (Dublin), sponsored by the Beer and Health Initiative (The Dutch Beer Institute foundation-The Brewers of Europe); all outside the submitted work. M Endres reports grants or contracts from Bayer as an unrestricted grant to Charite for MonDAFIS study and Berlin AFib registry; consulting fees from Bayer paid to their institution; payment or honoraria for lectures, presentations, speakers bureaus, manuscript writing or educational events from Bayer, Boehringer Ingelheim, Pfizer, Amgen, GSK, Sanofi, and Novartis, all paid to their institution; participation on a Data Safety Monitoring Board or Advisory Board with BMS as a Country PI for Axiomatic-SSP, Bayer as Country PO for NAVIGATE-ESUS, AstraZeneca, Boehringer Ingelheim, Daiichi Sankyo, Amgen, Covidien, with all fees paid to their institution; leadership or fiduciary role in board, society, committee or advocacy group, paid or unpaid with EAN as part of the Board of Directors, DGN, ISCBFM, AHA/ASA, ESO, WSO, DZHK (German Centre of Cardiovascular Research) as a PI, all of which are unpaid positions, and with DZNE (German Center of Neurodegenerative Diseases) as a paid PI; receipt of PCSK9 inhibitors for mouse studies from Amgen; all outside the submitted work. I Filip reports payment or honoraria for lectures, presentations, speakers bureaus, manuscript writing or educational events from Avicenna Medical and Clinical Research Institute, outside the submitted work. A Gialluisi reports grants or contracts from Italian Ministry of Economic Development (PLATONE project, bando "Agenda Digitale" PON I&C 2014–2020; Prog. n. F/080032/01-03/X35) paid to their institution, outside the submitted work. C Herteliu reports grants or contracts from Romanian National Authority for Scientific Research and Innovation, CNDS-UEFISCDI project number PN-III-P4-ID-PCCF-2016-0084 research grant (Oct 2018-Sept 2022) "Understanding and modelling time-space patterns of psychology-related inequalities and polarization," and project number PN-III-P2-2.1-SOL-2020-2-0351 research grant (June-Oct, 2021) "Approaches within public health management in the context of COVID-19 pandemic," and from the Ministry of Labour and Social Justice, Romania, project number 30/PSCD/2018 research grant (Sept 2018-June 2019) "Agenda for skills Romania 2020–2025;" all outside the submitted work. S Islam reports grants or contracts from National Heart Foundation Vanguard grant, Postdoctoral Fellowship and NHMRC Emerging Leadership Fellowship, outside the submitted work. Y Kalkonde reports grants or contracts from DBT/Wellcome Trust India Alliance as a DBT/Wellcome Trust India Alliance fellow in Public Health (grant number IA/CPHI/14/1/501514). M Kivimaki reports support for the present manuscript from The Wellcome Trust (221854/Z/20/Z) and Medical Research Council (MR/S011676/1), as research grants paid to their institute. K Krishnan reports non-financial support from UGC

Centre of Advanced Study, Phase II, Department of Anthropology, Panjab University, Chandigarh, India, outside the submitted work. P Lavados reports grants or contracts from Boehringer Ingelheim as grant support to their institute for RECCA stroke registry, and from ANID as personal grant support for ADDSPISE trial; payment or honoraria for lectures, presentations, speakers bureaus, manuscript writing or educational events from Boehringer Ingelheim as personal honoraria for lectures; support for attending meetings and/or travel from Boehringer Ingelheim as support for attending Global Stroke Netowrk meeting in 2020; all outside the submitted work. W Lo reports grants or contracts from 1U01NS106655-01A1 (MPI: S Ramey, Lo), 5U24NS1072050-02 (PI: Kolb), and 1P2CHD101912-01 (PD: S. Ramey), all outside the submitted work. S Lorkowski reports grants or contracts from Akcea Therapeutics Germany as payments made to their institution; consulting fees from Danone, Swedish Orphan Biovitrum (SOBI), and Upfield; payment or honoraria for lectures, presentations, speakers bureaus, manuscript writing or educational events from Akcea Therapeutics Germany, AMARIN Germany, Amedes Holding, AMGEN, Berlin-Chemie, Boehringer Ingelheim Pharma, Daiichi Sankyo Deutschland, Danone, Hubert Burda Media Holding, Lilly Deutschland, Novo Nordisk Pharma, Roche Pharma, Sanofi-Aventis, and SYNLAB Holding Deutschland & SYNLAB Akademie as personal payments; support for attending meetings and/or travel from Amgen as personal payments; participation on a Data Safety Monitoring Board or Advisory Board with Akcea Therapeutics Germany, Amgen, Daiichi Sankyo Deutschland, and Sanofi-Aventis as personal payments; all outside the submitted work. N Manafi reports support for the present manuscript from BMGF as funding to IHME for this project. B Norrving reports consulting fees from AstraZeneca and Bayer as personal payments outside the submitted work. O Odukoya reports grants or contracts from the Fogarty International Center of the National Institutes of Health as protected time towards the research reported in this publication was supported under the Award Number K43TW010704. The content is solely the responsibility of the authors and does not necessarily represent the official views of the National Institutes of Health. A Pana reports grants or contracts from Romanian National Authority for Scientific Research and Innovation, CNDS-UEFISCDI project number PN-III-P4-ID-PCCF-2016-0084 research grant (Oct 2018-Sept 2022) "Understanding and modelling time-space patterns of psychology-related inequalities and polarization," and project number PN-III-P2-2.1-SOL-2020-2-0351 research grant (June-Oct 2021) "Approaches within public health management in the context of COVID-19 pandemic," all outside the submitted work. M Postma reports leadership or fiduciary role in other board, society, committee or advocacy group, paid or unpaid with the UK's JCVI as an unpaid member, outside the current manuscript. A Radfar reports payment or honoraria for lectures, presentations, speakers' bureaus, manuscript writing or educational events from Avicenna Medical and Clinical Research Institute. S Sacco reports grants or contracts from Novartis and Allergan-AbbVie; consulting fees from Allergan-AbbVie, Novartis, Eli Lilly, AstraZeneca, and Novo Nordisk; payment or honoraria for lectures, presentations, speakers bureaus, manuscript writing or educational events from Allergan-AbbVie, Novartis, Eli Lilly, TEVA, Abbott, Medscape, and Olgology; support for attending meetings and/or travel from Allergan, Eli Lilly, Abbott, Novartis, and Teva; leadership or fiduciary role in other board, society, committee or advocacy group, paid or unpaid with Guideline Board European Stroke Organization as Co-chair, and with the European Headache Federation as a board member; all outside the submitted work. A Schutte reports payment or honoraria for lectures, presentations, speaker's bureaus, manuscript writing or educational events from Sanofi, Takeda, Abbott, Servier, and Omron Healthcare as honoraria for lectures during educational events; support for attending meetings and/or travel from Takeda and Omron; all outside the submitted work. J Singh reports consulting fees from Crealta/Horizon, Medisys, Fidia, Two labs, Adept Field Solutions, Clinical Care options, Clearview healthcare partners,

Putnam associates, Focus forward, Navigant consulting, Spherix, MedIQ, UBM LLC, Trio Health, Medscape, WebMD, and Practice Point communications; and the National Institutes of Health and the American College of Rheumatology; payment or honoraria for lectures, presentations, speakers bureaus, manuscript writing or educational events from Simply Speaking; support for attending meetings and/or travel from OMERACT, an international organization that develops measures for clinical trials and receives arm's length funding from 12 pharmaceutical companies, when traveling to OMERACT meetings; participation on a Data Safety Monitoring Board or Advisory Board as a member of the FDA Arthritis Advisory Committee; leadership or fiduciary role in other board, society, committee or advocacy group, paid or unpaid, with OMERACT as a member of the steering committee, with the Veterans Affairs Rheumatology Field Advisory Committee as a member, and with the UAB Cochrane Musculoskeletal Group Satellite Center on Network Meta-analysis as a director and editor; stock or stock options in TPT Global Tech, Vaxart pharmaceuticals, Charlotte's Web Holdings and previously owned stock options in Amarin, Viking, and Moderna pharmaceuticals; all outside the submitted work. S Stortecky reports grants or contracts from Edwards Lifesciences, Medtronic, Abbott, and Boston Scientific as grants made to their institution; consulting fees from Boston Scientific/BTG Teleflex; payment or honoraria for lectures, presentations, speakers bureaus, manuscript writing or educational events from Boston Scientific; all outside the submitted work. M Woodward reports consulting fees from Amgen as personal payment.

## Author Contributions

**Conceptualization:** Thanh G. Phan, Richard Beare, John Ly, Henry Ma.

**Data curation:** Philip M. Bath.

**Formal analysis:** Thanh G. Phan, Svitlana Ievlieva.

**Resources:** Stella Ho.

**Supervision:** Thanh G. Phan, Richard Beare.

**Writing – original draft:** Thanh G. Phan, Svitlana Ievlieva.

**Writing – review & editing:** Thanh G. Phan, Richard Beare, Philip M. Bath, Stella Ho, John Ly, Amanda G. Thrift, Velandai K. Srikanth, Henry Ma.

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
