## [Decision Letter · Decision Letter 0]

2 Oct 2022

PONE-D-22-20118Effect of alteplase, benzodiazepines and beta-blocker on post-stroke pneumonia: exploration of VISTA-AcutePLOS ONE

Dear Dr. Phan,

Thank you for submitting your manuscript to PLOS ONE. After careful consideration, we feel that it has merit but does not fully meet PLOS ONE’s publication criteria as it currently stands. Therefore, we invite you to submit a revised version of the manuscript that addresses the points raised during the review process.

Please, provide an accurate point-by-point reply to all the question raised by reviewers, and submit your revised manuscript by Nov 16 2022 11:59PM. If you will need more time than this to complete your revisions, please reply to this message or contact the journal office at plosone@plos.org. Please include the following items when submitting your revised manuscript:A rebuttal letter that responds to each point raised by the academic editor and reviewer(s). You should upload this letter as a separate file labeled 'Response to Reviewers'.A marked-up copy of your manuscript that highlights changes made to the original version. You should upload this as a separate file labeled 'Revised Manuscript with Track Changes'.An unmarked version of your revised paper without tracked changes. You should upload this as a separate file labeled 'Manuscript'.

We look forward to receiving your revised manuscript.

Kind regards,

Massimiliano Toscano

Academic Editor

PLOS ONE

Journal Requirements:

2. Thank you for including your ethics statement:  "The VISTA-Acute data were pooled from deidentified randomized control trials. Patients had to sign written consent to participate in the trials."  

For studies reporting research involving human participants, PLOS ONE requires authors to confirm that this specific study was reviewed and approved by an institutional review board (ethics committee) before the study began. Please provide the specific name of the ethics committee/IRB that approved your study, or explain why you did not seek approval in this case.

4. One of the noted authors is a group or consortium VISTA-Acute Collaborators. In addition to naming the author group, please list the individual authors and affiliations within this group in the acknowledgments section of your manuscript. Please also indicate clearly a lead author for this group along with a contact email address.

6. Please upload a copy of Supplementary Table 1 which you refer to in your text on page 11.

Reviewers' comments:

Reviewer's Responses to Questions

**Comments to the Author**

1. Is the manuscript technically sound, and do the data support the conclusions?

Reviewer #1: Yes

Reviewer #2: Yes

Reviewer #3: Yes

2. Has the statistical analysis been performed appropriately and rigorously? 

Reviewer #1: Yes

Reviewer #2: Yes

Reviewer #3: I Don't Know

3. Have the authors made all data underlying the findings in their manuscript fully available?

Reviewer #1: Yes

Reviewer #2: No

Reviewer #3: Yes

4. Is the manuscript presented in an intelligible fashion and written in standard English?

Reviewer #1: Yes

Reviewer #2: Yes

Reviewer #3: Yes

5. Review Comments to the Author

Reviewer #1: The paper by Phan et al. deals with the use of a multivariate, machine learning-based approach to find predictor of risk for post-stroke pneumonia. This paper falls into a large an interesting field and has some added value to the matter. The paper highlights the conundrum that curative interventions for stroke as alteplase has no impact on pneumonia, Similarly also thrombectomy didn’t provide a significant impact on pneumonia

However, I find some critical points that I think are worthy to be taken into account. I think that authors should discuss in more detail the role of NIHSS in the discussion. This result in fact falls in line with similar results that reinforced the validity of this new paper. In several studies NIHSS resulted as the stronger predictors, and also in a nomogram (Willeke F Westendorp et al Eur Stroke J. 2018 Jun;3(2):136-144) and recently confirmed by an independent group with a different statistical method (Tommaso B Jannini et al T B Jannini Neurol Sci. 2022 Feb; 43(2):1167-1176). In particular, the result that NIHSS equal or superior to 10 is a predictor of pneumonia is very interesting since Toscano et al Eur Neurol. 2015;74(3-4):171-7. Found that a similar NIHSS is predictive of persistent dysphagia, which is a risk factor for stroke related pneumonia.

In this line I think that the decision of not including dysphagia among predictors hampers the analysis, and it should be mentioned in the limitation section. As far as I know the field of multivariable statistics, these approaches tend to provide some data mining. They are very much influenced by the covariates that are chosen to be inserted. In particular, if big and strong predictors are excluded, small and less relevant covariates can appear as valid predictors in absence of a better parameters. In the paper by Jannini et al. 2022, the scale used to diagnose dysphagia (GLOBE vs WST) was predictive of the occurrence of pneumonia

I would recommend to add a line in which authors acknowledge that the absence of dysphagia as a predictor can modify the results but that the fact that drugs, with exception of BDZ, do not influenced pneumonia when analyzed without a strong predictor as dysphagia is a reinforcing aspect that medication has a small role in determine pneumonia. In the conclusion, since authors suggest more research, if they want, they could suggest to use a single validated scale to record dysphagia so that it can be used as covariate of multivariable analysis.

Reviewer #2: This paper from Phan et al. explored the effect of alteplase on post-stroke pneumonia occurrence. Collaterally, the effect of other drugs, such as benzodiazepines and beta-blockers, was explored. The study pointed out two interesting results: the first one is the neutral effect of alteplase on post-stroke pneumonia occurrence, and the second one is the protective effect of benzodiazepines on post-stroke pneumonia occurrence.

The statistical analysis seems to be sophisticated and difficult to catch by a non-expert.

In addition to all the limitations listed by the authors, my major concerns are about the methodology.

• First, the effect of alteplase is demonstrated not only by the low NIHSS after thrombolysis, which could simply reflect a low baseline NIHSS, but also by the delta NIHSS (24h NIHSS-baseline NIHSS). Therefore, more conveniently, the effect of alteplase could be unmasked by the delta NIHSS which really reflects the effect of alteplase (efficacy, reperfusion, no reflow…). Since this datum seems to have been collected by authors, the analysis should be re-run in this way to be reliable.

• This is a retrospective study of prospectively collected individual patient data (VISTA-Acute trial) and this could be counted as another limitation. Pneumonia has been assessed retrospectively extracting data on adverse events from a larger database and details on the type of dysphagia screen used are lacking. The management of patients could therefore be very heterogeneous among different centers limiting the significance of the effect of alteplase on post-stroke pneumonia occurrence. A possible way to minimize this heterogeneity could be to insert the variable ‘center’ in the analysis to unmask a possible ‘center-’ effect.

Page 3 line15, replace ‘pneumia’ with ‘pneumonia’

Page 3 line 22, replace ‘devised’ with ‘divided’

Abstract- lines2-3: the sentence does not make sense. It must be reworded.

Reviewer #3: Phan et al. prepared a manuscript aiming to investigate the “Effect of alteplase, benzodiazepines and beta-blocker on post-stroke pneumonia: exploration of VISTA-Acute”.

1. The abstract is concise explaining the work clearly and briefly.

2. Introduction: The strong relationship between stroke severity and pneumia. Change to pneumonia. VISTA, explain in the introduction.

3. Methods: I suggest the authors to analyze more the terms used for collecting pneumonia participants and if possible, the ICD-10 codes. Here in, the authors need to describe more about the groups of this study.

4. Outcomes: 5653 patients. Pneumonia frequency 614. Try to start with the highlight of this study.

5. Statistical analysis: Well detailed.

6. Discussion: The main findings in our study are that RTPA and beta-blockers had a minimal effect on the proportion of people having pneumonia following stroke. Is this in line with other studies? This finding on stroke severity and age is consistent with other studies on stroke associated pneumonia, references. “Our finding suggests a small effect of benzodiazepine on pneumonia”. One of the common complications in the acute phase in AIS patients is short/long- duration epileptic seizures causing pneumonia which usually can be treated with benzodiazepines. Patients with psychomotor anxiety also receive medication in acute phase.

6. PLOS authors have the option to publish the peer review history of their article (what does this mean?). If published, this will include your full peer review and any attached files.

Reviewer #1: No

Reviewer #2: No

Reviewer #3: No

---

## [Author Response · Author response to Decision Letter 0]

7 Jan 2023

Re: PONE-D-22-20118

Effect of alteplase, benzodiazepines and beta-blocker on post-stroke pneumonia: exploration of VISTA-Acute

We thank the journal for the chance to revise the paper. The reference format has been changed to reflect the journal’s style. The ethics statement has been amended. The data used in this study is available on request from VISTA-Acute who owns the data. The ethics statement has been moved to the Method. The supplementary Figure has been added.

Reviewer #1: The paper by Phan et al. deals with the use of a multivariate, machine learning-based approach to find predictor of risk for post-stroke pneumonia. This paper falls into a large an interesting field and has some added value to the matter. The paper highlights the conundrum that curative interventions for stroke as alteplase has no impact on pneumonia, Similarly also thrombectomy didn’t provide a significant impact on pneumonia

However, I find some critical points that I think are worthy to be taken into account. I think that authors should discuss in more detail the role of NIHSS in the discussion. This result in fact falls in line with similar results that reinforced the validity of this new paper. In several studies NIHSS resulted as the stronger predictors, and also in a nomogram (Willeke F Westendorp et al Eur Stroke J. 2018 Jun;3(2):136-144) and recently confirmed by an independent group with a different statistical method (Tommaso B Jannini et al T B Jannini Neurol Sci. 2022 Feb; 43(2):1167-1176). In particular, the result that NIHSS equal or superior to 10 is a predictor of pneumonia is very interesting since Toscano et al Eur Neurol. 2015;74(3-4):171-7. Found that a similar NIHSS is predictive of persistent dysphagia, which is a risk factor for stroke related pneumonia.

Thank you for the suggestions. We have included these new references as well as discussed importance of NIHSS in Discussion on pages 12.

In this line I think that the decision of not including dysphagia among predictors hampers the analysis, and it should be mentioned in the limitation section. As far as I know the field of multivariable statistics, these approaches tend to provide some data mining. They are very much influenced by the covariates that are chosen to be inserted. In particular, if big and strong predictors are excluded, small and less relevant covariates can appear as valid predictors in absence of a better parameters. In the paper by Jannini et al. 2022, the scale used to diagnose dysphagia (GLOBE vs WST) was predictive of the occurrence of pneumonia.

We did not have data on dysphagia. We have explored dysphagia in data from our own centre using Shapley value regression to determine the contribution of dysphagia and found it to be low relative to NIHSS. We do not see this as a potential issue as the random forest strategy is different from regression strategy which cannot handle large number of covariates. Furthermore, patients developed pneumonia even when kept nil by mouth. This limitation and the discussion has been added to the Discussion on page 12.

I would recommend to add a line in which authors acknowledge that the absence of dysphagia as a predictor can modify the results but that the fact that drugs, with exception of BDZ, do not influenced pneumonia when analyzed without a strong predictor as dysphagia is a reinforcing aspect that medication has a small role in determine pneumonia. In the conclusion, since authors suggest more research, if they want, they could suggest to use a single validated scale to record dysphagia so that it can be used as covariate of multivariable analysis.

We have added caveat to our model but because dysphagia could not be analyzed we cannot make strong statement about dysphagia in the VISTA dataset. This has been added in Discussion page 13

Reviewer #2: This paper from Phan et al. explored the effect of alteplase on post-stroke pneumonia occurrence. Collaterally, the effect of other drugs, such as benzodiazepines and beta-blockers, was explored. The study pointed out two interesting results: the first one is the neutral effect of alteplase on post-stroke pneumonia occurrence, and the second one is the protective effect of benzodiazepines on post-stroke pneumonia occurrence.

The statistical analysis seems to be sophisticated and difficult to catch by a non-expert. In addition to all the limitations listed by the authors, my major concerns are about the methodology.• First, the effect of alteplase is demonstrated not only by the low NIHSS after thrombolysis, which could simply reflect a low baseline NIHSS, but also by the delta NIHSS (24h NIHSS-baseline NIHSS). Therefore, more conveniently, the effect of alteplase could be unmasked by the delta NIHSS which really reflects the effect of alteplase (efficacy, reperfusion, no reflow…). Since this datum seems to have been collected by authors, the analysis should be re-run in this way to be reliable.

The data came from clinical trial registry and we cannot go back to check for other data. We are aware of the potential issue of change in NIHSS. This issue was related to the imbalance in the covariate. The frequency of pneumonia was 11.5 % in the RTPA (270/2358) and 10.4% (344/3295) in the no RTPA groups but there was significant (p<0.05) imbalance in covariates (age, baseline National Institutes of Health Stroke Scale (NIHSS), diabetes, and sex). This was discussed in the Discussion. We did propensity matching of the RTPA group (not presented here) but found no difference. This was an unexpected finding for us as it was the reason behind the study. I agree with the reviewer that we will acknowledge that the data may differ had we used data from RTPA trials. We will approach other investigators to perform this analysis. This has been added on page 14.

• This is a retrospective study of prospectively collected individual patient data (VISTA-Acute trial) and this could be counted as another limitation. Pneumonia has been assessed retrospectively extracting data on adverse events from a larger database and details on the type of dysphagia screen used are lacking. The management of patients could therefore be very heterogeneous among different centers limiting the significance of the effect of alteplase on post-stroke pneumonia occurrence. A possible way to minimize this heterogeneity could be to insert the variable ‘center’ in the analysis to unmask a possible ‘center-’ effect.

This is an excellent suggestion but the data came from a registry where multiple trials have been pooled together. It’s not possible to identify which trial the data came from. Data on centres are also not available. This aspect will be added to the Discussion under Limitation, pages 13-14

Page 3 line15, replace ‘pneumia’ with ‘pneumonia’

This change has been made.

Page 3 line 22, replace ‘devised’ with ‘divided’

This change has been made.

Abstract- lines2-3: the sentence does not make sense. It must be reworded.

This change has been made. “Investigators have described its associations with beta-blocker. However, there has been no evaluation of the role of recombinant tissue plasminogen activator (RTPA).”

Reviewer #3: Phan et al. prepared a manuscript aiming to investigate the “Effect of alteplase, benzodiazepines and beta-blocker on post-stroke pneumonia: exploration of VISTA-Acute”.

1. The abstract is concise explaining the work clearly and briefly.

2. Introduction: The strong relationship between stroke severity and pneumia. Change to pneumonia. VISTA, explain in the introduction.

This typo has been corrected. The abbreviation VISTA has been explained.

3. Methods: I suggest the authors to analyze more the terms used for collecting pneumonia participants and if possible, the ICD-10 codes. Here in, the authors need to describe more about the groups of this study.

The data came from VISTA. As such we were not able to analyze using ICD-10 codes. We have discussed this limitation on page 13.

4. Outcomes: 5653 patients. Pneumonia frequency 614. Try to start with the highlight of this study.

We have made changes to the abstract and Results on page 8 as suggested. 

5. Statistical analysis: Well detailed.

6. Discussion: The main findings in our study are that RTPA and beta-blockers had a minimal effect on the proportion of people having pneumonia following stroke. Is this in line with other studies? This finding on stroke severity and age is consistent with other studies on stroke associated pneumonia, references. “Our finding suggests a small effect of benzodiazepine on pneumonia”. One of the common complications in the acute phase in AIS patients is short/long- duration epileptic seizures causing pneumonia which usually can be treated with benzodiazepines. Patients with psychomotor anxiety also receive medication in acute phase.

This is an interesting hypothesis, that benzodiazepine use was related to seizure and which we were not able to test in this data. In our published data we had observed that pneumonia tend to occur relatively early, within the first 2 days after stroke onset. A previous analysis from VISTA (Sykora Stroke 2015) had proposed that beta-blocker may reduce the risk of pneumonia. In our study, we had evaluated the proportional contribution of beta-blocker to pneumonia. We have provided in Supplementary Figure 2 the variables which interact with beta-blockers (strongest with hypertension and RTPA). We have added this on page 14. In another analysis, Frank and colleagues did not observe an effect of benzodiazepine on pneumonia (Int J Stroke 2014). We were concerned with psychotropic medications and had explored this but found that they made very small contribution if at all.

Once again, we thank the Editor and reviewers for a chance to revise the paper. 

Yours sincerely,

Prof Thanh G Phan

---

## [Decision Letter · Decision Letter 1]

30 Jan 2023

Effect of alteplase, benzodiazepines and beta-blocker on post-stroke pneumonia: exploration of VISTA-Acute

PONE-D-22-20118R1

Dear Dr. Phan,

We’re pleased to inform you that your manuscript has been judged scientifically suitable for publication and will be formally accepted for publication once it meets all outstanding technical requirements.

Kind regards,

Massimiliano Toscano, M.D., PhD

Academic Editor

PLOS ONE

Additional Editor Comments (optional):

The authors followed all the recommendations given and answered all the doubts raised by modifying the text as required. The manuscript has been improved and, as far as I am concerned, the paper is now suitable for publication.

Reviewers' comments:

Reviewer's Responses to Questions

**Comments to the Author**

1. If the authors have adequately addressed your comments raised in a previous round of review and you feel that this manuscript is now acceptable for publication, you may indicate that here to bypass the “Comments to the Author” section, enter your conflict of interest statement in the “Confidential to Editor” section, and submit your "Accept" recommendation.

Reviewer #1: All comments have been addressed

Reviewer #2: All comments have been addressed

2. Is the manuscript technically sound, and do the data support the conclusions?

Reviewer #1: Yes

Reviewer #2: Yes

3. Has the statistical analysis been performed appropriately and rigorously? 

Reviewer #1: Yes

Reviewer #2: Yes

4. Have the authors made all data underlying the findings in their manuscript fully available?

Reviewer #1: Yes

Reviewer #2: No

5. Is the manuscript presented in an intelligible fashion and written in standard English?

Reviewer #1: Yes

Reviewer #2: Yes

6. Review Comments to the Author

Reviewer #1: Dear authors, you have fully addressed my comments. I endorse the publication of the present study.

Reviewer #2: (No Response)

7. PLOS authors have the option to publish the peer review history of their article (what does this mean?). If published, this will include your full peer review and any attached files.

Reviewer #1: No

Reviewer #2: No

---

## [Editor Report · Acceptance letter]

2 Feb 2023

PONE-D-22-20118R1 

Effect of alteplase, benzodiazepines and beta-blocker on post-stroke pneumonia: exploration of VISTA-Acute 

Dear Dr. Phan:

I'm pleased to inform you that your manuscript has been deemed suitable for publication in PLOS ONE. Congratulations! Your manuscript is now with our production department. 

Kind regards, 

on behalf of

Dr. Massimiliano Toscano 

Academic Editor

PLOS ONE